# Observational study of azithromycin in hospitalized patients with COVID-19

**Alejandro Rodríguez-Molinero**[1]*, **Carlos Pérez-López**[2], **César Gálvez-Barrón**[1], **Antonio Miñarro**[3], **Oscar Macho**[1], **Gabriela F. López**[1], **Maria Teresa Robles**[1], **María Dolores Dapena**[1], **Sergi Martínez**[1], **Ezequiel Rodríguez**[1], **Isabel Collado**[1], on behalf of the COVID-19 research group of CSAPG[¶]

**1** Research Area, Consorci Sanitari de l'Alt Penedès i Garraf (CSAPG), Sant Pere de Ribes, Barcelona, Spain, **2** Technical Research Center for Dependency Care and Autonomous Living (CETpD), Universitat Politècnica de Catalunya, Vilanova i la Geltru, Spain, **3** Department of Genetics, Microbiology and Statistics, School of Biology, University of Barcelona, Barcelona, Spain

¶ Membership of the COVID-19 research group of CSAPG is provided in the Acknowledgments.
* rodriguez.molinero@gmail.com

**Data Availability Statement:** Full data cannot be shared publicly because of the risk of re-identification of some patients included in the

## Abstract

### Background

The rapid spread of the disease caused by the novel SARS-CoV-2 virus has led to the use of multiple therapeutic agents whose efficacy has not been previously demonstrated. The objective of this study was to analyze whether there is an association between the use of azithromycin and the evolution of the pulmonary disease or the time to discharge, in patients hospitalized with COVID-19.

### Methods

This was an observational study on a cohort of 418 patients admitted to three regional hospitals in Catalonia, Spain. As primary outcomes, we studied the evolution of SAFI ratio (oxygen saturation/fraction of inspired oxygen) in the first 48 hours of treatment and the time to discharge. The results were compared between patients treated and untreated with the study drug through subcohort analyses matched for multiple clinical and prognostic factors, as well as through analysis of non-matched subcohorts, using Cox multivariate models adjusted for prognostic factors.

### Results

There were 239 patients treated with azithromycin. Of these, 29 patients treated with azithromycin could be matched with an equivalent number of control patients. In the analysis of these matched subcohorts, SAFI at 48h had no significant changes associated to the use of azithromycin, though azithromycin treatment was associated with a longer time to discharge (10.0 days vs 6.7 days; log rank: p = 0.039). However, in the unmatched cohorts, the increased hospital stay associated to azithromycin use, was no significant after adjustment using Multivariate Cox regression models: hazard ratio 1.45 (IC95%: 0.88–2.41; p = 0.150). This study is limited by its small sample size and its observational nature; despite the strong

database. Data are available from the Consorci Sanitari de l'Alt Penedès-Garraf Institutional Data Access (contact via mail: recerca@csapg.cat) for researchers who meet the criteria for access to confidential data.

**Funding:** The author(s) received no specific funding for this work.

**Competing interests:** The authors have declared that no competing interests exist.

pairing of the matched subcohorts and the adjustment of the Cox regression for multiple factors, the results may be affected by residual confusion.

## Conclusions

We did not find a clinical benefit associated with the use of azithromycin, in terms of lung function 48 hours after treatment or length of hospital stay.

## Introduction

In December 2019, an epidemic outbreak associated with a novel coronavirus (severe acute respiratory syndrome coronavirus 2, SARS-CoV-2) was reported in Wuhan (China) with mainly respiratory clinical manifestations [1]. The extent of the outbreak reached such a magnitude that the WHO declared it a pandemic on March 12, 2020 [2]. Although mortality rates in those affected (approximately 2% among medically treated patients) [3] seem to be overestimated due to underdiagnosis of affected individuals with mild symptoms, the extent of the pandemic has caused the search for effective treatments to become a top priority.

Several pharmacological agents have been proposed as potential treatments based on theoretical considerations, in vitro studies, or clinical trials conducted in conditions caused by related viruses [4–6]. However, current evidence has not confirmed the presence or absence of a benefit of these treatments and even warns of the probable risks or adverse effects associated with their use. Several randomized clinical trials are underway but have not yet been completed or have not been reported [7, 8].

Azithromycin has been considered, usually combined with hydroxychloroquine, based on its in vitro action against other viruses, such as influenza A [9, 10], and its potential immuno-modulatory and anti-inflammatory action in other respiratory diseases [11–13]. Randomized clinical trials specifically evaluating the clinical benefit of azithromycin, as an isolated treatment for COVID-19, are in course [14], however, their results have yet not been reported. From observational studies, at the moment this manuscript is being written, only Rosenberg et al. [15] and Arshad et al. [16] have reported results on the specific use of azithromycin in Covid-19 disease. In both studies, among hospitalized patients (n = 1428 and n = 2541, respectively) treated or not with azithromicyn and/or hydroxychloroquine, a benefit on mortality was not found in the group treated with azithromicyn alone. Other studies have evaluated the use of azithromycin, though in combination with hydroxychloroquine [17–21]. In a randomized, open-label and controlled trial by Cavalcanti et al. (n = 667) [20], a benefit on clinical severity was not found in the group of hospitalized patients treated with the combination of hydroxychloroquine plus azithromycin. There are other studies on the combination but with results that are difficult to interpret because they lacked a comparison group [17, 18, 22] or because some of the patients assigned to the treated group received only hydroxychloroquine and not the combination with azithromycin [23, 24].

The absence of an adjusted comparison group in the majority of reported observational studies is relevant given the well-known confounding of the observational design by factors that influence the choice or not of a certain treatment. One of the methods used to balance this has been the propensity score [25], as used in the studies by Magagnoli et al. [19] and Geleris et al. [24], in which no benefit was found from the use of hydroxychloroquine (with or without azithromycin) in patients with COVID-19. However, this method is not the most appropriate when trying to obtain comparison groups that also match at the time, or follow-up time, in which certain factors appear (e.g., time when a third drug is introduced).

Therefore, we set out to analyze the relevant clinical parameters under the use of azithromycin in patients hospitalized in our health centers, through comparison of groups using multivariate analysis and matching techniques based on brute-force algorithms, which can match patients by variables that change over time.

## Method

This observational study was carried out on a cohort of 418 patients admitted to the hospitals of the Consorci Sanitari de l'Alt Penedès and Garraf (CSAPG), which includes three regional hospitals with a total of 275 acute-care beds and which serve a reference population of 247,357 inhabitants from the regions of Alt Penedès and Garraf, Catalonia, Spain.

Data were collected from all patients with a clinical picture compatible with COVID-19 (patients diagnosed of COVID-19 pulmonary disease by their doctors upon admission) seen between March 12 and May 2, 2020, from the time of admission to discharge or up to a maximum of 30 days after admission. Real-time reverse transcription polymerase chain reaction (RT -PCR) for SARS-CoV-2 was performed on a sample obtained by nasopharyngeal smear to all patients. Patients with negative RT-PCR test were excluded.

The data were collected from the electronic medical records by the COVID-19 research group of CSAPG. The data collected included sociodemographic data, previous diseases, chronic treatments, symptoms of disease presentation, vital signs, and clinical evolution each day since admission, including the need for oxygen therapy, the inspired fraction of oxygen (FiO2), and the oxygen administration system (nasal prongs, Venturi mask, reservoir mask, or invasive or noninvasive mechanical ventilation). All treatments given during admission were recorded, as well as all analyses and chest radiographs performed. The researchers responsible for data collection collected the data using a structured form created in the OpenClinica, version 3.1. (Copyright © OpenClinica LLC and collaborators, Waltham, MA, USA), following a common procedure on which they were previously trained. Quality controls were established during the data collection process, and the errors detected were corrected; the responsible researchers were retrained when necessary.

As exposure variable, treatment with azithromycin was considered. Azithromycin, according to the hospital protocol, was prescribed at a dose of 500 mg on the first day (oral or intravenous), followed by 250 mg daily, until completing 5 days of treatment. A patient was considered exposed to azithromycin if they received at least three doses of the drug. The main outcome variables for the efficacy analyses were time to discharge and oxygen saturation (%)/FiO$_2$(%) ratio (SAFI) at 48 hours after the start of treatment [26, 27]. As secondary variables, SAFI in the first 96 hours after treatment, and mortality were analyzed.

In the statistical analysis, a double strategy was used: 1) analysis of subcohorts paired by confounding factors and 2) analysis of unpaired subcohorts, adjusted for confounding factors.

As part of the first strategy, a subcohort of patients treated with the study drug was formed, and a control subcohort was matched with the treatment group (1:1 match ratio). The patients were matched by the following prognostic markers, which were collected dichotomously (Yes/No) after detailed reading of all available patient reports: sex, age, obesity, heart failure, chronic renal failure, and sleep apnea–hypopnea syndrome (SAHS).

The above listed prognostic markers, which were used as matching criteria, were identified in multivariate binary logistic regression models, in which severe disease (defined as need for oxygen therapy with a non-rebreathing masks or mechanical ventilation) and death were taken as dependent variables. The variables introduced in these models were pre-selected from those pathological antecedents with significant association to the outcomes (bivariate analyses; p<0.05) by using the Lasso technique [28]. Virtually all the available pathological history of

the patients was tested: cardiovascular, digestive, osteoarticular, pulmonary, endocrine, neurological, psychiatric diseases, kidney failure, neoplasms, autoimmune diseases and several immunodeficiencies (all the diseases within these categories were treated in a dichotomic way: presence vs absence of the disease).

Follow-up of each patient started the day the patient took the first dose of a study drug. Follow-up of each control started the day after admission on which SAFI, vital signs (blood pressure and heart rate), radiological involvement, and C-reactive protein (CRP), were similar to those of the patient with whom they were matched. For this purpose, the CRP on day 1 of follow-up of the patient or, failing that, the day before the start of treatment, was taken as reference. Likewise, the radiological involvement on the treatment started, or any previous time up to a maximum of two days before the start of treatment, was considered. Missing data on radiological involvement were imputed in the following way: It was assumed that the radiological involvement on the days between two equal radiographs was the same as on the days of said radiographs (e.g., if a patient had an X-ray with three affected quadrants on day 1 and another with three affected quadrants on day 6, it was assumed that on all intervening days they had three affected quadrants). This interpolation was allowed up to a maximum interval of 6 days between radiographs. No missing data were imputed for other variables. Patients who received the study treatment and their controls were matched only if they had received the same other treatments for COVID-19, including hydroxychloroquine, lopinavir/ritonavir, interferon, corticosteroids, or tocilizumab. A margin of 3 days of lag at the start of the other treatments was tolerated between the patients under study and the matched controls. In preliminary analyzes, we found no effect of heparin treatment on the results of this study, which is why this drug was not included among the matching criteria.

For pairing, a first step was performed using brute-force computing algorithms, which identified all possible controls in the database for each of the patients who received the study treatment. In this first step, controls were chosen who had the same sex and state of obesity ("yes" vs "no", according to the clinical history), the same radiological involvement (number of affected quadrants on anteroposterior radiography: 0–4) and an age difference not exceeding 15 years. The control was allowed to have a SAFI from 1.1 points lower to 2 points higher than the treated patient and a CRP from 6 mg/dL lower to 4 mg/dL higher than the treated patient. The matching was then refined, choosing from among the previously identified potential controls the most similar in terms of SAFI, blood pressure, heart rate, and CRP by the propensity score. The complete process of selecting patient pairs, including the procedures performed by the brute force algorithms, are summarized in Fig 1.

The success of the matching was verified by comparing means or percentages between groups. A different trend in the evolution of patients (improvement in one group and worsening in another) was discarded, verifying that the difference between the SAFI was similar between day 1 of analysis and the day before entering the analysis. In the matched subcohorts, the SAFI was studied at 48, 72, and 96 hours using Student's t-test for independent samples and the time to discharge using the log-rank test. In the SAFI analyses, patients with palliative sedation were excluded because in these patients SAFI is not related to the severity of the disease. In the analysis of time to discharge, deceased patients were excluded.

In the analysis of unmatched subcohorts (second analysis strategy), the effect of azithromycin was analyzed in a subcohort in which all patients had been treated with hydroxychloroquine and lopinavir/ritonavir, from which patients treated with corticosteroids or other drugs that were distributed significantly asymmetrically between groups were excluded. In the analysis of these subcohorts, the total length of hospital stay was counted from day 1 of admission, and patients were considered exposed to the study drug if they had taken it at any time since admission (at least three doses). The time to discharge (excluding deceased patients) and

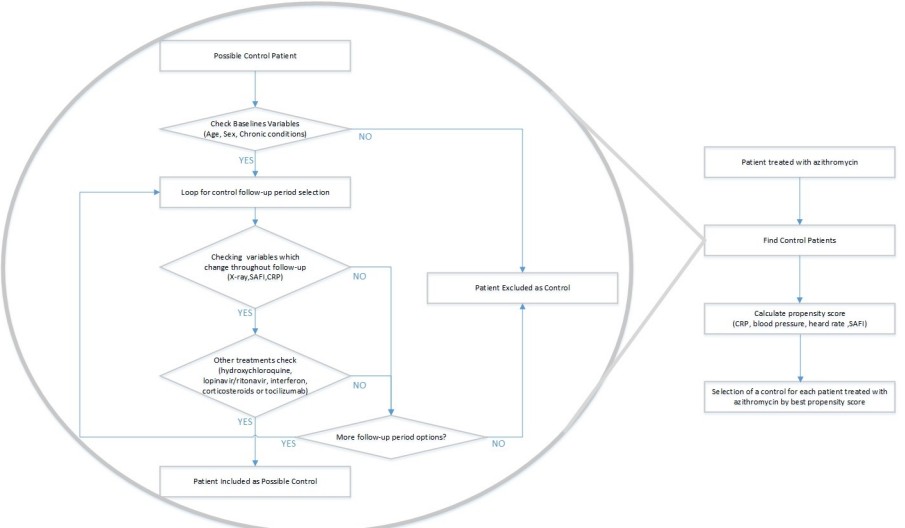

**Fig 1. Process of selection of matched controls.**

mortality were studied by fitting Cox regression models adjusted for the following covariates: sex, age, obesity, heart failure, chronic renal failure, SAHS, baseline saturation in the emergency room, CRP in the emergency room, and quadrants affected in the emergency radiography.

For the statistical analysis, R software version 3.6.1 (R Project for Statistical Computing) and IBM SPSS statistics version 26 were used.

The research ethics committee Bellvitge Hospital reviewed the study and accepted the waiver of the patient's informed consent, as it was an observational and ambispective review of clinical data, and the patient's personal data were anonymized for its publication. Approval from the Ethical Committee was granted before starting data collection.

## Results

Of the 464 consecutive patients with a clinical diagnosis of COVID-19 and pulmonary involvement who were admitted between March 12 and May 2, 2020, 46 were excluded for having a negative RT-PCR for SARS-CoV-2. Of the 418 patients included in the analysis, 238 (56.9%) were men and 180 (43.1%) were women, the mean age of the sample was 65.4 years (SD 16.6 years), and the median follow-up was 8 days (IQR 5–12 days). In total, 239 (57.2%) patients were treated with azithromycin. Patients who were treated with both hydroxychloroquine and lopinavir/ritonavir during admission totaled 346 (82.8%). In the first 30 days after admission, 79 patients died (18.9%). Fig 2 shows a flow diagram of the sample of the study.

The characteristics of the matched subcohorts are shown in Table 1. Comparing to the source cohort, matched cohorts had a larger proportion of men, and included patients with better respiratory function (higher saturation and less chest-x-ray involvement) as can be seen in Table 2. The characteristics of the unmatched subcohorts are shown in Table 3. Table 4 shows the mean change in saturation, $FiO_2$ and SAFI, with respect to baseline, after 48, 72, and 96 hours of treatment, in the matched subcohorts. In the analysis of matched cohorts, hospital satay was significantly longer in patients treated with azithromycin, compared with their paired controls (Logrank; p = 0.039). However, in the unmatched cohorts, the increased hospital stay associated to azithromycin use, was no significant after adjustment using Multivariate

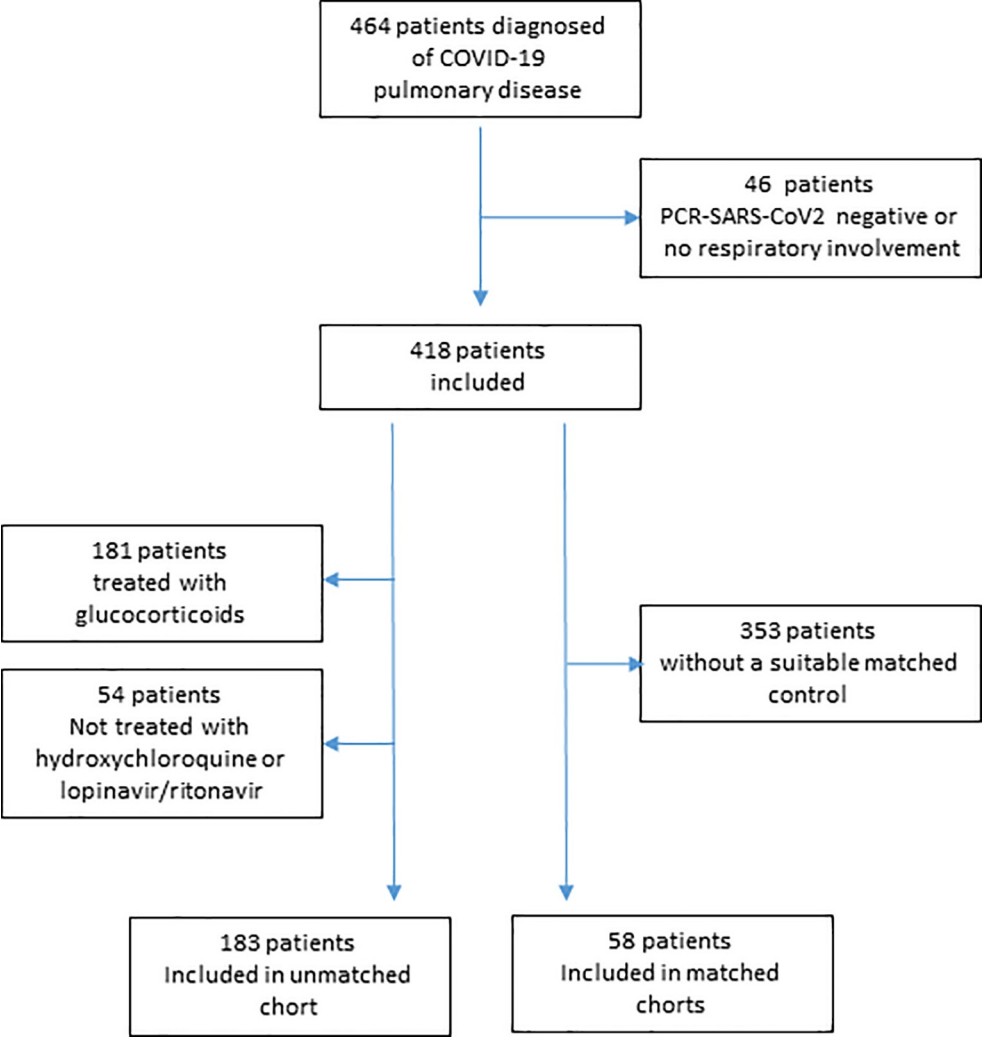

**Fig 2. Flow diagram.**

Cox regression models: hazard ratio 1.45 (IC95%: 0.88–2.41; p = 0.150). Fig 3 shows the unadjusted Kaplan-Meier comparison curves and log-rank test for the time to discharge in all studied subcohorts.

Six deaths (8.1%) were recorded in the unmatched control subcohort (treated with hydroxychloroquine and lopinavir/ritonavir) and 7 people (5.3%) died in the subcohort receiving additional treatment with azithromycin (p = 0.501). In the matched subcohorts 3 people died, 1 of them (3.4%) in the azithromycin group, and 2 (6.9%) in the control group. We considered this numbers of events insufficient to draw conclusions.

## Discussion

Our study did not find a benefit associated with the use of azithromycin in terms of respiratory function (SAFI), or time to discharge. In fact, hospital stay was longer in the azithromycin treated group, compared with matched controls.

Most of the patients in our study received hydroxychloroquine and lopinavir/ritonavir, therefore our results are mainly related to the potential benefit of adding azithromycin to this

**Table 1.  Baseline characteristics of patients treated with azithromycin and their matched controls.**

|  | Azithromycin (n29) | Control (n29) | p |
|---|---|---|---|
| **Age** (years) | 63.0 | 63.1 | 0.987 |
| **Men** (n,%) | 21 (72.4%) | 21 (72.4%) | 1.000 |
| **Obesity** (n) | 3 (10.3%) | 3 (10.3%) | 1.000 |
| **CHF** (n) | 1 (3.5%) | 1 (3.5%) | 1.000 |
| **CRF** (n) | 2 (6.7%) | 5 (17.2%) | 0.423 |
| **SAHS** (n) | 3 (10.3%) | 3 (10.3%) | 1.000 |
| **Tobacco** (n) | 1 (3.4%) | 3 (10.3%) | 0.611 |
| **Hypertension** (n) | 12 (41.4%) | 11 (37.9%) | 1.000 |
| **Diabetes** (n) | 5 (20.7%) | 6 (17.2%) | 1.000 |
| **COPD** (n) | 1 (3.4%) | 1 (3.4%) | 1.000 |
| **Other cardiopathy** (n) | 5 (17.2%) | 2 (6.9%) | 0.433 |
| **Saturation** (%) | 96.1 | 96.4 | 0.521 |
| **Systolic BP** (mmHg) | 121.6 | 123.4 | 0.690 |
| **Diastolic BP** (mmHg) | 70.7 | 68.7 | 0.484 |
| **HR** (bpm) | 77.8 | 78.5 | 0.783 |
| **Temperature** (˚C) | 36.7 | 36.6 | 0.686 |
| **SAFI**[1] | 3.6 | 3.5 | 0.668 |
| **SAFI trend**[2] | 0.0 | 0.0 | 0.983 |
| **Radiographic involvement**[3] | 1.5 | 1.5 | 0.632 |
| **CRP** (mg/dL) | 7.4 | 8.0 | 0.655 |
| **Urea** (mg/dL) | 41.3 (n23)* | 37.7 (n19)* | 0.649 |
| **Neutrophils** (10e9/L) | 4.3 (n25)* | 5.4 (n6)* | 0.302 |
| **Lymphocytes** (10e9/L) | 1.1 (n25)* | 1.0 (n6)* | 0.511 |
| **Hydroxychloroquine** (n) | 27.0 | 26.0 | 0.640 |
| **Lop/Rit** (n) | 27.0 | 26.0 | 0.640 |
| **Interferon** (n) | 3 | 4 | 0.687 |
| **Tocilizumab** (n) | 5 | 3 | 0.706 |
| **Methylprednisolone** (n) | 1 | 0 | 0.313 |
| **Dexamethasone** (n) | 8 | 7 | 0.764 |
| **Hospital stay** (days) | 10.0 | 6.7 | 0.025 |

CHF: congestive heart failure. CRF: chronic renal failure. SAHS: sleep apnea–hypopnea syndrome. BP: blood pressure. HR: heart rate. SAFI: saturation (%)/fraction of inspired $O_2$ (%). CRP: C-reactive protein.

[1] Maximum value 4.76, corresponding to 100% saturation with $FiO_2$ of 21%.

[2] Change in SAFI with respect to the day before the start of the follow-up period.

[3] Number of affected quadrants in an anteroposterior chest radiograph. Range: 0–4 (0: no involvement; 4: involvement of the upper and lower lobes of both lungs).

* Information not available for all the patients.

drug regimen. Our results are in line with the work by Cavalcanty et al. [20] who found no benefit on clinical severity in the group of patients receiving hydroxychloroquine plus azithromycin compared to the group who received only hydroxychloroquine. Rosenberg et al. [15] and Arshad et al. [16] investigated similar patients in terms of setting (hospitalized patients), severity (moderate or severe disease) and oxygenation parameters and found no benefit on hospital mortality in the group of patients treated with azithromycin alone.

Our sample included only hospitalized patients, so our results should be considered in the realm of hospital management of COVID-19 and cannot be extrapolated to patients with mild symptoms in whom outpatient treatment and monitoring is usually recommended. In

**Table 2. Comparison between matched subcohorts and source cohort.**

|  | Total cohort (n 418) | Matched subcohorts (n 58) | p |
|---|---|---|---|
| **Age** | 65,4 | 63.1 | 0.320 |
| **Men** | 238 (57,1%) | 42 (72.4%) | 0.032 |
| **Obesity** | 74 (17.7%) | 6 (10.3%) | 0.192 |
| **CHF** | 26 (6,2%) | 2 (3,4%) | 0.558 |
| **CRF** | 61 (14,6%) | 7 (12,1%) | 0.693 |
| **SAHS** | 34 (8,1%) | 6 (10,3%) | 0.611 |
| **Tobacco** | 36 (8,6%) | 4 (6,9%) | 0,804 |
| **Hypertension** | 217 (51,9%) | 23 (39.7%) | 0.093 |
| **Diabetes** | 99 (23,7%) | 11 (19.0%) | 0,268 |
| **COPD** | 41 (9.8%) | 2 (3.4%) | 0.143 |
| **Other cardiopathy** | 62 (14.8%) | 7 (12.1%) | 0.693 |
| **Saturation** | 91,6 | 93.9 | 0.031 |
| **Radiographic involvement** [1] | 2,07 | 1,59 | 0,003 |
| **CRP (mg/dL)** | 12.4 (n156)* | 9.8 | 0,231 |
| **Urea (mg/dL)** | 48.0 (n337)* | 38.3 (n42)* | 0.088 |
| **Neutrophils (10e9/L)** | 6.0 | 4.7 | 0.015 |
| **Lymphocytes (10e9/L)** | 1.1 | 1.1 | 0.654 |
| **Hospital stay** | 9.3 | 9.2 | 0.775 |

* Information is not available for all the patients.

[1] Number of affected quadrants in an anteroposterior chest radiograph. Range: 0–4 (0: no involvement; 4: involvement of the upper and lower lobes of both lungs).

addition, matching criteria cause patient selection, so that only those patients for whom a pair is found, enter the analysis (possibly selecting the most frequent type of patient). This means that the selected subcohorts do not represent well the total hospital population with COVID-19 and the results of this study are only applicable to patients with features similar to ours.

Given the observational nature of this study, the existence of residual confounders cannot be ruled out, thus, a possibility exists that patients assigned to azithromycin treatment would have a higher-risk factors or disease severity. In the case of azithromycin, we believe that this problem is not likely since its use was widespread and not related to the severity of the disease. In any event, the exhaustive matching method used and the verification of the comparability of the groups lead us to assume that this confounding effect was unlikely and, if there, was small. Some factors could affect hospital discharge beyond the resolution of the infection, such as the presence of complications or factors related to social circumstances, especially in elderly patients (which could have prevented them from returning home). Since all the patients in the sample were admitted for COVID-19, the complications derived from hospitalization for this disease, seem to us to be part of the clinical picture we are studying and, therefore, their influence in the outcome seems appropriate. Regarding social factors affecting discharge, we think that the age matching of the paired subcohorts, should have mitigated its possible influence in the results. In any event, the most important confounder that could have affected the time to discharge is death, which was appropriately controlled, by excluding the deceased patients from this analysis.

The worse respiratory function at 72 hours of treatment, observed in azithromycin matched subcohort, may be affected by a selection bias, as there was a loss of data in four matched controls at the time of this comparison, which was a secondary endpoint (loss of patients may have unbalance the previously matched groups).

**Table 3. Baseline characteristics of the subcohorts of patients treated with hydroxychloroquine/lopinavir-ritonavir and patients with additional treatment with azithromycin (un-matched subcohorts).**

|  | HCL/LOP (n 63) | HCL/LOP/AZT (n 120) | p |
|---|---|---|---|
| **Age** | 57.2 | 61.6 | 0.066 |
| **Men** | 35 (52.2%) | 57 (47.5%) | 0.534 |
| **Obesity** | 12 (17.6%) | 17 (14.2%) | 0.526 |
| **CHF** | 3 (4.4%) | 7 (5.8%) | 1.000 |
| **CRF** | 4 (5.9%) | 14 (11.7%) | 0.195 |
| **SAHS** | 5 (7.4%) | 10 (8.3%) | 0.812 |
| **Tobacco** | 3 (4.4%) | 17 (14.2%) | 0,028 |
| **Hypertension** | 24 (35.3%) | 57 (47.5%) | 0.126 |
| **Diabetes** | 11 (16.2%) | 25 (20.8%) | 0,563 |
| **COPD** | 1 (1.5%) | 7 (5.8%) | 0,262 |
| **Other cardiopathy** | 9 (13,2%) | 16 (13.3%) | 1.000 |
| **Saturation** | 93.7 | 94.4 | 0.301 |
| **Radiographic involvement** [1] | 2.13 | 1.68 | 0.004 |
| **CRP (mg/dL)** | 9.9 (n26)* | 8.2 (n33)* | 0.295 |
| **Urea (mg/dL)** | 33.2 (n 38)* | 37.6 (n 111)* | 0.440 |
| **Neutrophils (10e9/L)** | 5.1 (n38)* | 5.0 (n115)* | 0.942 |
| **Lymphocytes (10e9/L)** | 1.2 (n38)* | 1.2 (n115)* | 0.732 |
| **Hydroxychloroquine** | 63 (100%) | 120 (100%) | - |
| **Lop/Rit** | 63 (100%) | 120 (100%) | - |
| **Interferon** | 12 (17.6%) | 12 (10.0%) | 0.131 |
| **Tocilizumab** | 7 (10.3%) | 11 (9.2%) | 0.801 |
| **Methylprednisolone** | 0 | 0 | - |
| **Dexamethasone** | 0 | 0 | - |
| **Hospital stay** (days) | 5.7 | 8.5 | <0.001 |

AZT: Azithromycin. CHF: congestive heart failure. CRF: chronic renal failure. CRP: C-reactive protein. HCQ: hydroxychloroquine. L/R: lopinavir/ritonavir. SAHS: sleep apnea-hypopnea syndrome.

[1] Number of affected quadrants in an anteroposterior chest radiograph. Range: 0–4 (0: no involvement; 4: involvement of the upper and lower lobes of both lungs).

* Information not available for all the patients.

**Table 4. Change in respiratory function parameters with respect to the first day of follow-up in patients treated with azithromycin.**

|  | Azithromycin | Control | Mean difference (IC95%) | p |
|---|---|---|---|---|
| **Saturation increment** |  |  |  |  |
| 48 hours | -0.82 (n29) | -0.81 (n29) | 0.02 (-1.35; 1.39) | 0.980 |
| 72 hours | -0.58 (n29) | -0.43 (n25) | 0.15 (-1.18; 1.48) | 0.821 |
| 96 hours | -0.91 (n27) | -0.40 (n24) | 0.51 (-0.72; 1.74) | 0.411 |
| **FiO$_2$ increment** |  |  |  |  |
| 48 hours | 4.93 (n29) | 3.33 (n29) | -1.60 (-11.36;8.16) | 0.744 |
| 72 hours | 9.07 (n29) | 0.56 (n27) | -8.51 (-21.77; 4.75) | 0.203 |
| 96 hours | 6.65 (n27) | -5.45 (n23) | -12.10 (-23.70; -0.50) | **0.041** |
| **SAFI increment** |  |  |  |  |
| 48 hours | -0.19 (n29) | -0.01 (n29) | 0.19 (-0.26; 0.64) | 0.408 |
| 72 hours | -0.23 (n29) | 0.34 (n25) | 0.57 (0.01; 1.14) | **0.046** |
| 96 hours | -0.08 (n27) | 0.40 (n23) | 0.49 (-0.08; 1.05) | 0.074 |

FiO$_2$: fraction of inspired oxygen.

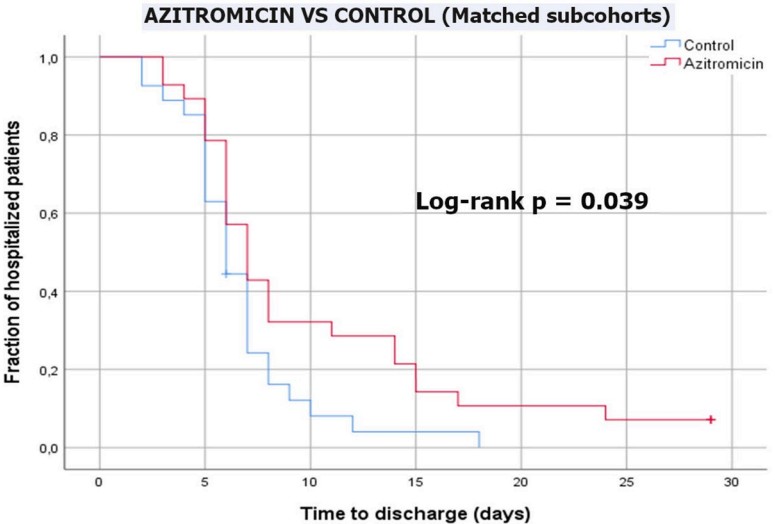

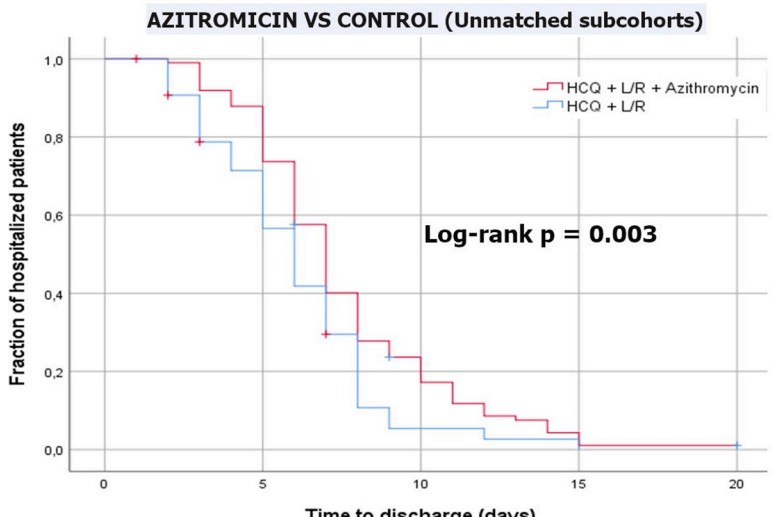

**Fig 3. Kaplan-Meier comparison curves and log-rank test outcomes of the different subcohorts (unadjusted).**

Our study was limited by its small sample size, which caused problems of statistical power, especially in the case of some of the outcomes of greatest interest, such as mortality, which cannot be sufficiently studied in this sample. In addition, the use of secondary data obtained from the clinical history might have led to information biases. However, given that the main variables were quantitative parameters that were little influenced by the observers or their expertise in measurement, and given that these parameters are routinely collected in clinical practice and hospital management, we consider unlikely the existence of a relevant bias of this type. In any case, the sample size and observational nature of our study make it necessary to wait for the results of randomized clinical trials, which are ongoing, to confirm ours.

In conclusion, in this observational study, we did not find evidence of a clinical benefit from the use of azithromycin in patients hospitalized with COVID-19. The use of azithromycin may be associated with worse clinical results, compared with matched controls.

## Acknowledgments

This research was conducted by the **COVID-19 research group of CSAPG** (led by Alejandro Rodríguez-Molinero: e-mail: rodriguez.molinero@gmail.com), which includes, in addition to the authors of this papers: Alberti Casas, Anna PhD, MD; Avalos Garcia, Jose L MD; Borrego Ruiz, Manel BS; Añaños Carrasco, Gemma MD; Campo Pisa, Pedro L; Capielo Fornerino, Ana M. MD; Chamero Pastilla, Antonio MD; Fenollosa Artés, Andreu MD; Gris Ambros, Clara MD; Hernandez Martinez, Lourdes MD; Hidalgo García, Antonio MD; Martín Puig, Mireia MD; Milà Ràfols, Núria RN; Molina Hinojosa, José C. MD; Monaco, Ernesto E MD; Peramiquel Fonollosa, Laura MD; Pisani Zambrano, Italo G. MD; Rives, Juan P. MD; Sabria Bach, Enric. MD; Sanchez Rodriguez, Yris M. MD; Segura Martin, Maria del Mar. RN; Tremosa Llurba, Gemma MD; Ventosa Gili, Ester MD; Venturini Cabanellas, Florencia I. MD; Vidal Meler, Natalia. MD. Group affiliations: Àrea de Recerca, Consorci Sanitari de l'Alt Penedès i Garraf, Vilafranca del Penedès, Barcelona (Spain).

We would like to thank Gloria Moes for her invaluable help in coordinating the fieldwork. Gloria Alba, Nuria Pola and Anna María Soler, for their initial help in collecting drug data. Montserrat Pérez and Rosa Guilera, for their help with the electronic medical record, and to David Blancas and Lourdes Gabarró for their work in the hospital protocols for COVID-19, and their initial supply of bibliography. We also should thank the CSAPG informatics team, for their technical support during the study. Finally, we should thank to the manager of the Consorci Sanitari de l'Alt Penedès i Garraf, José Luis Ibáñez Pardos, and the management team, for making this study possible.

## Author Contributions

**Conceptualization:** Alejandro Rodríguez-Molinero, César Gálvez-Barrón.

**Data curation:** Carlos Pérez-López, Oscar Macho, Gabriela F. López, Maria Teresa Robles, María Dolores Dapena, Sergi Martínez, Ezequiel Rodríguez, Isabel Collado.

**Formal analysis:** Carlos Pérez-López, Antonio Miñarro.

**Investigation:** Alejandro Rodríguez-Molinero.

**Methodology:** Alejandro Rodríguez-Molinero, César Gálvez-Barrón.

**Project administration:** Alejandro Rodríguez-Molinero.

**Supervision:** Alejandro Rodríguez-Molinero.

**Writing – original draft:** Alejandro Rodríguez-Molinero, César Gálvez-Barrón.

**Writing – review & editing:** Carlos Pérez-López, Antonio Miñarro, Oscar Macho, Gabriela F. López, Maria Teresa Robles, María Dolores Dapena, Sergi Martínez, Ezequiel Rodríguez, Isabel Collado.

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
