## [Decision Letter · Decision Letter 0]

22 Jul 2020

PONE-D-20-19615

Observational study of azithromycin in hospitalized patients with COVID-19

PLOS ONE

Dear Dr. Rodríguez-Molinero,

Thank you for submitting your manuscript to PLOS ONE. After careful consideration, we feel that it has merit but does not fully meet PLOS ONE’s publication criteria as it currently stands. Therefore, we invite you to submit a revised version of the manuscript that addresses the points raised during the review process.

The three reviewers found your study quite interesting but simultaneously raised some concerns which are mainly attributed to methodological issues and the way that results were interpreted. 

We look forward to receiving your revised manuscript.

Kind regards,

Stelios Loukides

Academic Editor

PLOS ONE

Journal Requirements:

2. Please ensure you have mentioned the potential limitations of this study also in your Abstract. Moreover, please ensure that no statements of causation, which cannot be supported by the observational nature of this study, are reported. Furthermore, please provide more detail on how matching was performed, and on the definitions used (for example, please define "heart failure"); and provide a citation for the algorithm used, or describe the methodology in more detail if not previously published.

3.We note that you have indicated that data from this study are available upon request. PLOS only allows data to be available upon request if there are legal or ethical restrictions on sharing data publicly. For information on unacceptable data access restrictions, please see http://journals.plos.org/plosone/s/data-availability#loc-unacceptable-data-access-restrictions.

4. One of the noted authors is a group or consortium [COVID-19 research group of CSAPG]. In addition to naming the author group, please list the individual authors and affiliations within this group in the acknowledgments section of your manuscript. Please also indicate clearly a lead author for this group along with a contact email address.

Reviewers' comments:

Reviewer's Responses to Questions

**Comments to the Author**

1. Is the manuscript technically sound, and do the data support the conclusions?

Reviewer #1: Partly

Reviewer #2: No

Reviewer #3: Yes

2. Has the statistical analysis been performed appropriately and rigorously? 

Reviewer #1: I Don't Know

Reviewer #2: No

Reviewer #3: I Don't Know

3. Have the authors made all data underlying the findings in their manuscript fully available?

Reviewer #1: No

Reviewer #2: No

Reviewer #3: No

4. Is the manuscript presented in an intelligible fashion and written in standard English?

Reviewer #1: Yes

Reviewer #2: Yes

Reviewer #3: Yes

5. Review Comments to the Author

Reviewer #1: This is an interesting paper on an important topic with patient data drawn from a huge database. There is value in the study.

But, the value seems a bit hidden to me. First, even though I am an academic clinician who has cared for lots of sick people, I have never used the SAFI measurement. This leaves me uncertain of the value of the findings related to SAFI, and the validity of this as a measure of progression of pulmonary findings is never discussed. In addition, without knowing weaning parameters, it is not clear if the use of oxygen is based only on pulmonary disease or if oxygen could have been continued because the patient was still sick with other issues. Second, the basis of the matching procedure is not clear. It is surprising that out of 239 azithromycin-treated patients and a similar number of un-treated patients, there would only be 29 pairs that were similar enough to compare. I'd need to know more about the matching procedures to know if the selected 29 patients/pairs were really representative of the larger population of COVID-19-positive patients.

The ethics statement is reasonable, but the actual paper only says that ethical approval was requested, not granted.

The data statement is confusing. It is not clear what "re-identification" of patients means and why de-identified data could not be made available.

Abstract. It would help in the Background to say what was being tested (such as severity of pulmonary disease) instead of the obscure (to me) ratio that is of uncertain clinical relevance.

Abstract. The results are only about a small part of the results presented in the paper. And, by mixing the analyses without fully explaining them in the abstract, it sounds contradictory - azithromycin was associated with a longer time to discharge but was also of "no significant difference." It is also not clear why one would look at a log rank of a simple measure such as length of stay.

The third paragraph of the introduction could be updated with new studies when a revised manuscript is submitted.

The first mention of the matching is incomplete. Only later do we learn that obesity was a yes/no characterization, but we still don't hear how much obesity counted or how obesity was defined. Matching age to within 15 years seems a bit broad - the risks of bad outcomes are reported to be very different between 60 and 75 year olds, for instance.

The SAFI data in the abstract are only for 48 hours, but there were significant differences (p<0.05) in Fi)2 and SAFI increment at 72 and 96 hours. It is not clear why one time point being "not different" makes it into the final conclusion of the study, and the other "significant" findings do not. The "loss of data" explanation does not generate much confidence in the conclusions.

It would help to mention a p value with the 8.1% vs 5.3% mortality figures.

These seem to be valid and important data in this study, but attention to these points would help at least me better understand the meaning and significance of some of the reported details. Thanks!

Reviewer #2: The methodology used in the paperi is interesting. The authors have tried to solve the issue of the absence of a control group and to reduce the impact of confounding factors by using matching techniques based on brute-force algorithms to identify a subcohort of patients paired by confounding factors and a second subcohort of patients where confounding factors were adjusted according to a normal distribution.

However there al many major issues which should be addressend in the manuscript.

Methods

To identify the paired subcohort if patients authors have matched the initial sample according to prognostic markers identified at a bivariate analysis and a multivariate model. Could you please explicit which variables have you initially considered for the bivariate analysis and the multivariate model and which significativity threshold have you used?

Could you please add a flow diagram (according to the STROBE guidelines) summarizing how from the source population the two subcohorts have been identified?

One of the pairing criteria were other concomitant treatments for COVID-19, but among those you have considered there isn’t heparin. Could you please motivate this choice?

There are some concernings regarding the use of lenght of hospital stay as a primary endpoint. Could you please discuss if there are other factors which can affect the leght of hospital stay besides the infection resolution?

Results

In the paired cohort, 72% of patients were male, while in the source population authors report a prevalence of male subjects of 56,9%. The subcohort of paired patients does not represents anymore the population sample. An analysis evaluating differences between the paired subcohort and the source population should be performed and summarized in a table showing significativity.

Table 1: Please add variables percentages consistently with the format of table 2

According to table 1, 27 (or 26) out of 29 patients in the paired subcohort, have been treated with hydroxychloroquine and lopinavir/ritonavir with (or without) aziththromycin. So you are not evaluating the efficacy of azythromycin in COVID-19, but the effect of the addittion of azithromycin to the combination of hydroxychloroquine and lopinavir/ritonavir. Please consider this in your discussion.

Population characteristics in table 1 and 2 should be better described. Please add smoke hystory, hypertension, diabetes, COPD, cardiovascular disease other than CHF, WBC and lenght of hospital stay.

It is not clear how many patients died in which subcohort and group. Furthermore authors says that numbers of deaths are insuffient to draw conclusions but few lines below they say “Our study found no benefit associated with the use of azithromycin in terms of respiratory function (SAFI), time to discharge, or mortality”. This is a conclusion regarding mortality.

Please express all results with 95%IC

Reviewer #3: Title: Observational study of azithromycin in hospitalized patients with COVID-19

This study aimed to analyze whether there is an association between the use of azithromycin and the evolution of saturation/fraction of inspired oxygen (SAFI) or time to discharge in patients hospitalized with COVID-19 in a Spanish cohort.

Comment 1: T I recommend checking the text "covid-19" and change it to "COVID-19”.

Comment 2: In the methods section, the authors define that they included patients with a clinical picture compatible with COVID-19. What were these criteria (clinical, imaging, nucleic acid amplification tests, serology)? I recommend adding it to the section.

Comment 3: Include in the methods section, it is not clear the type of diagnostic tests used, please detail it.

Comment 4: In the results section, the interquartile range of the median follow-up does not have one of the quartiles. Please verify this result.

6. PLOS authors have the option to publish the peer review history of their article (what does this mean?). If published, this will include your full peer review and any attached files.

Reviewer #1: No

Reviewer #2: No

Reviewer #3: **Yes: **Luis Gabriel Parra-Lara

---

## [Author Response · Author response to Decision Letter 0]

1 Aug 2020

Reviewer #1: 

1.This is an interesting paper on an important topic with patient data drawn from a huge database. There is value in the study.

But, the value seems a bit hidden to me. First, even though I am an academic clinician who has cared for lots of sick people, I have never used the SAFI measurement. This leaves me uncertain of the value of the findings related to SAFI, and the validity of this as a measure of progression of pulmonary findings is never discussed. In addition, without knowing weaning parameters, it is not clear if the use of oxygen is based only on pulmonary disease or if oxygen could have been continued because the patient was still sick with other issues. 

ANSWER: Oxygen saturation / FiO2 ratio (SAFI) is a parameter we choose, in order to make the oxygen saturation proportional to the amount of oxygen that it is being administered, otherwise, the oxygen saturation is not interpretable. SAFI (also known as S/F) is highly related to PAFI, which is Oxygen partial pressure / FiO2, and has been used, in order to interpret oxygen saturation in patients with oxygen therapy. 

Here we show you some examples from bibliography, were the SpO2/FIO2 ratio is used. We have now included in the paper a couple of them.

• Rice TW, et al. Comparison of the SpO2/FIO2 ratio and the PaO2/FIO2 ratio in patients with acute lung injury or ARDS. Chest. 2007;132(2):410-417. doi:10.1378/chest.07-0617

• Lu X, et al. Continuously available ratio of SpO2/FiO2 serves as a noninvasive prognostic marker for intensive care patients with COVID-19. Respir Res. 2020 Jul 22;21(1):194.

• Fernández-Ruiz M et al. for the treatment of adult patients with severe COVID-19 pneumonia: A single-center cohort study. J Med Virol. 2020 Jul 16; 

• Serpa Neto A et al. The use of the pulse oximetric saturation/fraction of inspired oxygen ratio for risk stratification of patients with severe sepsis and septic shock. J Crit Care. 2013 Oct;28(5):681-6.

• Schmickl CN et al. Decision support tool for early differential diagnosis of acute lung injury and cardiogenic pulmonary edema in medical critically ill patients. Chest. 2012 Jan;141(1):43-50. doi: 10.1378/chest.11-1496. Epub 2011 Oct 26. 

• Lobete Prieto C, et al. [Prediction of PaO₂/FiO₂ ratio from SpO₂/FiO₂ ratio adjusted by transcutaneous CO₂ measurement in critically ill children]. An Pediatr (Barc). 2011 Feb;74(2):91-6. 

• Tripathi RS et al. Pulse oximetry saturation to fraction inspired oxygen ratio as a measure of hypoxia under general anesthesia and the influence of positive end-expiratory pressure. J Crit Care. 2010 Sep;25(3):542.e9-13.

Honestly we consider that oxygen has not been continued for any other reason than the pulmonary disease (the whole database includes patients with COVID-19 pulmonary disease). In line with the reviewer’s comment, we took the precaution of excluding patients with palliative sedation, since in them oxygen therapy does not usually correspond to lung function.

Please see new references 26, 27 and 10th paragraph of the method section. 

2. Second, the basis of the matching procedure is not clear. It is surprising that out of 239 azithromycin-treated patients and a similar number of un-treated patients, there would only be 29 pairs that were similar enough to compare. I'd need to know more about the matching procedures to know if the selected 29 patients/pairs were really representative of the larger population of COVID-19-positive patients.

ANSWER: We intended to ensure that both groups were comparable. Thus, we matched for a large number of characteristics, including the many different treatments used for the disease and their timing within the admission. It is very difficult to find in a “natural cohort” (no inclusion criteria), pairs of patients so similar in so many respects. As physicians tend to use different treatments in different types of patients, it is hard to find totally equally pairs. That is why the size of the paired cohorts is small (small but comparable)

The reviewer is right when says that the matched cohort is not representative of the hospital population with COIVD-19. Paired cohorts closely resemble a clinical trial sample with strict inclusion criteria. Clinical trials often do not represent all patients with a certain disease, but rather the proportion that meets the trial criteria (which is sometimes a small proportion of the total population). The characteristics of the trial population are usually shown in publications’ table 1, which is equivalent to our table 1. The table helps to understand the type of patient’s to whom the results can be extrapolated. We have added a new table (table 2) which compares the paired cohorts and the source cohort. We have also commented on the differences in the results section. We have also discussed this topic in the discussion section, inviting readers not to generalize the results to any COVID patient, but only to those who resemble those included in our study.

Regarding the control selection process, we have added a new figure, explaining the procedure (new fig.1)

Please see changes in the 2nd paragraph of the results, 3rd paragraph of the discussion, table 2, and figure 1.

3. The ethics statement is reasonable, but the actual paper only says that ethical approval was requested, not granted.

ANSWER: The ethics approval was granted; we have changed the sentence to better express this fact. 

Please, see changes in the last paragraph of the methods section.

4. The data statement is confusing. It is not clear what "re-identification" of patients means and why de-identified data could not be made available.

ANSWER: As we gathered so many data of each single patient, some of them are still identifiable, even without de identification data (example: there are no so many people admitted to our hospital, who are male, 96 years old, had sleep apnea, and survived COVID-19, between March and May 2019). For this reason, we could not make data available, specially, when the patients did not consent for this study, as stated in the ethical approval. 

We could only publish a limited set of variables, aggregated (for example, in ranges of age of 5-10 years), previously revised and explicitly approved by our Ethical Committee. 

The editorial team has also raised the same question; we are answering in parallel to them. They will modify the statement, according to the information provided by us, and the final decision. 

5. Abstract. It would help in the Background to say what was being tested (such as severity of pulmonary disease) instead of the obscure (to me) ratio that is of uncertain clinical relevance.

ANSWER: We have added the suggestion of the reviewer to the abstract.

Please, see changes in the first paragraph of the abstract.

6. Abstract. The results are only about a small part of the results presented in the paper. And, by mixing the analyses without fully explaining them in the abstract, it sounds contradictory - azithromycin was associated with a longer time to discharge but was also of "no significant difference." It is also not clear why one would look at a log rank of a simple measure such as length of stay.

ANSWER: We have modified the abstract to make it more understandable, and also have added means of hospital stay for each groups.

Please, see changes in the third paragraph of the abstract.

7. The third paragraph of the introduction could be updated with new studies when a revised manuscript is submitted.

ANSWER: We have updated references to existing bibliography in this paragraph.

Please, see changes in the third paragraph of the introduction, and the second paragraph of the discussion.

8. The first mention of the matching is incomplete. Only later do we learn that obesity was a yes/no characterization, but we still don't hear how much obesity counted or how obesity was defined. Matching age to within 15 years seems a bit broad - the risks of bad outcomes are reported to be very different between 60 and 75 year olds, for instance.

ANSWER: This is a study based on secondary data (clinical records), which limits the use of strict definitions for patient characteristics. In the case of obesity, a patient was considered obese if, in the pathological history of his emergency report or admission note, a doctor had mentioned obesity as part of the chronic conditions description.

Regarding age, it must be considered that fifteen years is the maximum deviation allowed by the matching algorithm, the difference is not so large in most pairs. Given the high number of matched variables, a reduction in the age-matching margin would have limited the number of patients matched. In any event, existing age differences in patient-pairs, seems to be balanced (equally in favor and against the treatment group), as the mean age of groups are almost identical, which nullifies any possible effect of age.

We have tried to explain better the matching process (a new figure is now added), the selection of matching criteria, and how variables were collected. Pease see 6th and 7th paragraph of the methods, and the new figure 1.

9. The SAFI data in the abstract are only for 48 hours, but there were significant differences (p<0.05) in Fi)2 and SAFI increment at 72 and 96 hours. It is not clear why one time point being "not different" makes it into the final conclusion of the study, and the other "significant" findings do not. The "loss of data" explanation does not generate much confidence in the conclusions.

ANSWER: As it is shown in table 3 (now table 4), loss of data (reduction in n) affects only to variables after 72 hours. As the loss of patients’ data is not homogeneous between groups, and occurs after the moment of pairing, it is possible that it could lead to residual confusion (due to "un-pairing"). That it is why we were cautious, and we did not highlight this finding in the conclusions.

In any case, we have tested now, how the pairing of the patients remains after the losses regarding SAFI at 72 hours, and the groups continue to be well matched except for a greater presence of chronic renal failure before admission, in the azithromycin group (5 patients vs 1 patient). As this is not very likely to affect respiratory function on the third day of treatment, we have dared to add to the conclusions that “azithromycin may be associated with worse results.”

Please, see changes in the last paragraph of the discussion. 

10. It would help to mention a p value with the 8.1% vs 5.3% mortality figures.

ANSWER: We have included a p value for this difference. 

Please see the last paragraph of the results.

11. These seem to be valid and important data in this study, but attention to these points would help at least me better understand the meaning and significance of some of the reported details. Thanks!

ANSWER: We have tried to address each point raised by the reviewer. Many thanks for the careful review. 

Reviewer #2: 

1. The methodology used in the paperi is interesting. The authors have tried to solve the issue of the absence of a control group and to reduce the impact of confounding factors by using matching techniques based on brute-force algorithms to identify a subcohort of patients paired by confounding factors and a second subcohort of patients where confounding factors were adjusted according to a normal distribution.

However there al many major issues which should be addressend in the manuscript.

Methods

To identify the paired subcohort if patients authors have matched the initial sample according to prognostic markers identified at a bivariate analysis and a multivariate model. Could you please explicit which variables have you initially considered for the bivariate analysis and the multivariate model and which significativity threshold have you used?

ANSWER: We have explained now in the paper the bivariate analysis and multivariable models. Variables included in the models were pre-selected using the Lasso technique [reference 28], from those pathological antecedents, with statistical significance (p<0.05) in a bivariate analysis performed with all the known pathologies of the patients. Although p<0.05 may seem to be a too strict entry criterion for multivariable models, we did not consider it to be so, in the light of the very high number of comparisons performed in the bivariate analysis (p<0.05 was not corrected for the large number of comparisons in this step)

In response to the reviewer’s comment, we have now introduced a short explanation in the manuscript. Please see the 7th paragraph of the methods section. 

2. Could you please add a flow diagram (according to the STROBE guidelines) summarizing how from the source population the two subcohorts have been identified?

ANSWER: According to the reviewer’s comment, we have included a flow diagram. 

Please see new figure 2.

3. One of the pairing criteria were other concomitant treatments for COVID-19, but among those you have considered there isn’t heparin. Could you please motivate this choice?

ANSWER: The vast majority of patients in the sample had heparin, some of them at prophylactic doses, others at higher doses. We previously analyzed the effect of heparin on the outcomes of our study and found no effect, so we did not match the sample for this drug.

We have now added a sentence explaining it to the manuscript. Please see the 8th paragraph of the method section.

4. There are some concernings regarding the use of lenght of hospital stay as a primary endpoint. Could you please discuss if there are other factors which can affect the leght of hospital stay besides the infection resolution?

ANSWER: Other than the infection resolution, the main factor that can affect hospital stay is death; however, we excluded deceased patients from the analyses of time to discharge. 

Following the reviewer’s comment, we have included a discussion on other factors in the manuscript. Please see the 4th paragraph of the discussion section. 

5. In the paired cohort, 72% of patients were male, while in the source population authors report a prevalence of male subjects of 56,9%. The subcohort of paired patients does not represents anymore the population sample. An analysis evaluating differences between the paired subcohort and the source population should be performed and summarized in a table showing significativity.

ANSWER: We have added table comparing our matched cohorts with the source cohort, and commented the differences in the text.

Nevertheless, our paired subcohorts are not intendent to represent the population, they are intendent to be comparable, in order to test azithromycin efficacy. Just like in clinical trials, where the sample selected by the inclusion criteria typically does not represent all the population with a given disease. As a consequence, the readers of a clinical trial paper, should refer to the “table 1”, in order to verify the type of patients involved in the trial (ie. disease severity, age range…), because the results will only be applicable to patients with similar characteristics to those included (it is a well-known drawback of most clinical trials; for instance, excluding elderly people) Our matched study is a clinical trial-like research. Matching criteria are thought to make groups comparable, not to select a representative sample. Our Table 1 shows the characteristic of the matched patients, and readers should pay attention to this table, before generalizing conclusions to their COVID-19 patients.

We have added a comment regarding the “representativeness” of the matched cohorts in the discussion.

Please see the new table 2, changes in the second paragraph of the results and changes in the 3rd paragraph of the discussion.

6. Table 1: Please add variables percentages consistently with the format of table 2

ANSWER: Done. Please see changes in the table 3 (old table 2).

7. According to table 1, 27 (or 26) out of 29 patients in the paired subcohort, have been treated with hydroxychloroquine and lopinavir/ritonavir with (or without) aziththromycin. So you are not evaluating the efficacy of azythromycin in COVID-19, but the effect of the addittion of azithromycin to the combination of hydroxychloroquine and lopinavir/ritonavir. Please consider this in your discussion.

ANSWER: You are wright. We have tried to clarify this point. Please see changes in the second paragraph of the discussion.

8. Population characteristics in table 1 and 2 should be better described. Please add smoke hystory, hypertension, diabetes, COPD, cardiovascular disease other than CHF, WBC and lenght of hospital stay.

ANSWER: We have added to the tables, the characteristics requested by the reviewer.

Please see the new tables of the paper.

9. It is not clear how many patients died in which subcohort and group. Furthermore authors says that numbers of deaths are insuffient to draw conclusions but few lines below they say “Our study found no benefit associated with the use of azithromycin in terms of respiratory function (SAFI), time to discharge, or mortality”. This is a conclusion regarding mortality.

ANSWER: Thanks for the last comment, it is a mistake we did not detected in the first version. We have corrected accordingly the first paragraph of the discussion. In addition, we have changed the last paragraph of the results, to clarify the other issue raised by the reviewer and reports deaths in every subcohort. 

Please see corrections in the las paragraph of the results section and the first paragraph of the discussion section. 

10. Please express all results with 95%IC

ANSWER: We have added 95% IC to the results in the text and table 4.

We appreciate very much the careful review, which in our view have contributed to improve the manuscript. 

Reviewer #3: Title: Observational study of azithromycin in hospitalized patients with COVID-19

This study aimed to analyze whether there is an association between the use of azithromycin and the evolution of saturation/fraction of inspired oxygen (SAFI) or time to discharge in patients hospitalized with COVID-19 in a Spanish cohort.

Comment 1: T I recommend checking the text "covid-19" and change it to "COVID-19”.

ANSWER: We have updated the text following the reviewer suggestion.

Comment 2: In the methods section, the authors define that they included patients with a clinical picture compatible with COVID-19. What were these criteria (clinical, imaging, nucleic acid amplification tests, serology)? I recommend adding it to the section.

ANSWER: There were no defined set of clinical criteria. We included all patients whose doctors considered the COVID-19 as the clinical diagnosis upon admission, and who were treated with SARS-CoV-2 directed treatments (some of this patients were afterwards excluded when PCR results were available)

In line with the reviewer comment we have made a slight modification in the text, to clarify this point. Please see changes in the second paragraph of the methodology.

Comment 3: Include in the methods section, it is not clear the type of diagnostic tests used, please detail it.

ANSWER: We have now clarified the diagnostic test in the methods. Please see changes in the second paragraph of the methods section.

Comment 4: In the results section, the interquartile range of the median follow-up does not have one of the quartiles. Please verify this result.

We intended to show the “range” of the interquartile range rather than its limits. We have now display the interquartile range limits, for clarity (thanks, it is now more understandable). We also have detected an erratum in the median previously reported, which we have now corrected. 

Please see changes in the first paragraph of the results section.

We would like to thank reviewer 3 for his comments.

---

## [Decision Letter · Decision Letter 1]

24 Aug 2020

Observational study of azithromycin in hospitalized patients with COVID-19

PONE-D-20-19615R1

Dear Dr. Rodríguez-Molinero,

We’re pleased to inform you that your manuscript has been judged scientifically suitable for publication and will be formally accepted for publication once it meets all outstanding technical requirements.

Kind regards,

Stelios Loukides

Academic Editor

PLOS ONE

Additional Editor Comments (optional):

Reviewers' comments:

Reviewer's Responses to Questions

**Comments to the Author**

1. If the authors have adequately addressed your comments raised in a previous round of review and you feel that this manuscript is now acceptable for publication, you may indicate that here to bypass the “Comments to the Author” section, enter your conflict of interest statement in the “Confidential to Editor” section, and submit your "Accept" recommendation.

Reviewer #1: All comments have been addressed

Reviewer #3: All comments have been addressed

2. Is the manuscript technically sound, and do the data support the conclusions?

Reviewer #1: Yes

Reviewer #3: Yes

3. Has the statistical analysis been performed appropriately and rigorously? 

Reviewer #1: Yes

Reviewer #3: Yes

4. Have the authors made all data underlying the findings in their manuscript fully available?

Reviewer #1: Yes

Reviewer #3: No

5. Is the manuscript presented in an intelligible fashion and written in standard English?

Reviewer #1: Yes

Reviewer #3: Yes

6. Review Comments to the Author

Reviewer #1: (No Response)

Reviewer #3: (No Response)

7. PLOS authors have the option to publish the peer review history of their article (what does this mean?). If published, this will include your full peer review and any attached files.

Reviewer #1: No

Reviewer #3: **Yes: **Luis Gabriel Parra-Lara

---

## [Editor Report · Acceptance letter]

27 Aug 2020

PONE-D-20-19615R1 

Observational study of azithromycin in hospitalized patients with COVID-19 

Dear Dr. Rodríguez-Molinero:

I'm pleased to inform you that your manuscript has been deemed suitable for publication in PLOS ONE. Congratulations! Your manuscript is now with our production department. 

Kind regards, 

on behalf of

Dr Stelios Loukides 

Academic Editor

PLOS ONE